# Assessment of potential factors associated with the sensitivity and specificity of Sofia Influenza A+B Fluorescent Immunoassay in an ambulatory care setting

**Cristalyne Bell**[1‡]*, **Maureen Goss**[1‡], **Jennifer Birstler**[2‡], **Emily Temte**[1☉], **Guanhua Chen**[2☉], **Peter Shult**[3☉], **Erik Reisdorf**[3☉], **Thomas Haupt**[4☉], **Shari Barlow**[1☉], **Jonathan Temte**[1‡]

1 Department of Family Medicine and Community Health, School of Medicine and Public Health, University of Wisconsin, Madison, Wisconsin, United States of America, 2 Department of Biostatistics and Medical Informatics, School of Medicine and Public Health, University of Wisconsin, Madison, Wisconsin, United States of America, 3 Communicable Disease Division, Wisconsin State Laboratory of Hygiene, Madison, Wisconsin, United States of America, 4 Bureau of Communicable Diseases, Wisconsin Division of Public Health, Madison, Wisconsin, United States of America

☉ These authors contributed equally to this work.
‡ These authors contributed equally to this work and are considered the senior authors
* cristalyne.bell@fammed.wisc.edu

**Data Availability Statement:** Replication Data for: Assessment of potential factors associated with the sensitivity and specificity of Sofia Influenza A+B

## Abstract

### Background

Seasonal influenza leads to an increase in outpatient clinic visits. Timely, accurate, and affordable testing could facilitate improved treatment outcomes. Rapid influenza diagnostic tests (RIDTs) provide results in as little as 15 minutes and are relatively inexpensive, but have reduced sensitivity when compared to RT-PCR. The contributions of multiple factors related to test performance are not well defined for ambulatory care settings. We assessed clinical and laboratory factors that may affect the sensitivity and specificity of Sofia Influenza A+B Fluorescence Immunoassay.

### Study design

We performed a post-hoc assessment of surveillance data amassed over seven years from five primary care clinics. We analyzed 4,475 paired RIDT and RT-PCR results from specimens collected from patients presenting with respiratory symptoms and examined eleven potential factors with additional sub-categories that could affect RIDT sensitivity.

### Results

In an unadjusted analysis, greater sensitivity was associated with the presence of an influenza-like illness (ILI), no other virus detected, no seasonal influenza vaccination, younger age, lower cycle threshold value, fewer days since illness onset, nasal discharge, stuffy nose, and fever. After adjustment, presence of an ILI, younger age, fewer days from onset, no co-detection, and presence of a nasal discharge maintained significance.

Fluorescent Immunoassay in an ambulatory care setting are available through the Harvard Dataverse database (Dataset Persistent ID doi:10.7910/DVN/JNC1XJ.) https://dataverse.harvard.edu/dataset.xhtml?persistentId=doi:10.7910/DVN/JNC1XJ.

**Funding:** This study was unfunded, but utilized data collected for ongoing influenza surveillance that was made possible by grant funding to Wisconsin Department of Health Services by the Centers for Disease Control and Prevention and the Council of State and Territorial Epidemiologists.

**Competing interests:** JLT has received past research funding from Quidel Corporation. Quidel provided in-kind Sofia analyzers and Influenza A+B FIA tests to the Wisconsin surveillance team. This does not alter our adherence to PLOS ONE policies on sharing data and materials. Quidel did not direct or exert any influence over study design, data collection and analysis, decision to publish, or preparation of the manuscript.

## Conclusion

Clinical and laboratory factors may affect RIDT sensitivity. Identifying potential factors during point-of-care testing could aid clinicians in appropriately interpreting negative influenza RIDT results.

## Introduction

Seasonal influenza poses a significant annual disease burden [1–3] and is common in outpatient clinical settings. Although clinicians often diagnose possible influenza based on patient history and symptoms, studies indicate symptoms alone perform inadequately for influenza [4,5]. Rapid influenza diagnostic tests (RIDTs) have been shown to significantly improve the accuracy of physicians' estimates of influenza during peak influenza season [6]. RIDTs allow laboratory confirmation of clinically-suspected influenza cases within a timeframe that is clinically meaningful in primary care and urgent care settings and therapeutically meaningful for prompt initiation of antiviral therapy and avoidance of inappropriate antibiotic prescribing [7].

Point-of-care RIDTs are easy to use, relatively inexpensive, and provide results in as little as 15 minutes. Most RIDTs are highly specific (>95%), but they exhibit varying and often low sensitivity [8–10] when compared to reverse transcription–polymerase chain reaction (RT-PCR). Thus, the Center for Disease Control and Prevention (CDC) advises caution when interpreting negative test results [11].

Clinicians are rarely provided adequate training on the performance characteristics of RIDTs and approaches for an informed interpretation of results. Accordingly, falsely negative results can lead to missed opportunities or errors in treatment and patient education; falsely positive results may lead to inappropriate treatment.

Understanding which factors influence sensitivity may enable clinicians to better select appropriate patients for testing and improve their ability to interpret RIDT results. Pragmatic RIDT operating characteristics, however, have not been well defined in primary care settings. Studies that attempt to identify potential factors often limit the number of assessed variables. Several studies examined only two potential factors [6,12–14]. Age and virus strain were assessed most frequently [6,12–19]. Other factors included timing of illness onset relative to testing and illness severity [16–18], specimen collection method [19], and viral load or influenza RT-PCR cycle threshold (Ct) value [16,20–22]. To our knowledge, no study has examined the effect of within-season influenza vaccination status on sensitivity, or combined multiple clinically-relevant factors simultaneously.

We performed a post-hoc assessment of a large surveillance dataset to evaluate multiple clinical and laboratory factors that may affect RIDT sensitivity using seven years of data amassed from five primary care clinics that sequentially employed Sofia® Influenza A+B Immunoassay (Sofia-FIA; Quidel Corporation) for point-of-care influenza testing. We focused primarily on factors that clinicians could take into consideration during a clinical encounter. Our *a priori* hypotheses were that sensitivity declined with increasing age and with increasing time from illness onset. We were also interested in the combined roles of sex, influenza vaccination status, influenza type, meeting influenza-like illness (ILI) criteria, and influenza RT-PCR Ct value.

## Methods

The University of Wisconsin Health Sciences Minimal Risk Institutional Review Board deemed protocols exempt and classified the project as clinical care and public health

surveillance. Thus, patients were not required to sign a consent form to be enrolled in the surveillance program.

## Setting

Clinicians—including physicians, resident physicians, nurse practitioners, and physician assistants—at five primary care clinics in southcentral Wisconsin collected respiratory specimens on patients presenting with acute respiratory infections (ARIs) between October 26, 2012 and June 30, 2019. Four of the sites are University of Wisconsin Department of Family Medicine and Community Health residency training clinics. The fifth non-residency community clinic was incorporated into the surveillance program prior to the 2014–2015 influenza season. The clinics are located in two urban, one suburban, and two rural communities and serve diverse populations. All clinics were enrolled in the Influenza Incidence Surveillance Project (IISP), later renamed the Optional Influenza Surveillance Enhancement (OISE) program. IISP/OISE monitors medically attended influenza-like illness (ILI) and estimates the incidence of influenza [1,2]. The Centers for Disease Control and Prevention initiated IISP in 2009 and the program continues to operate year around. This platform—as implemented by the Wisconsin study team—provided an opportunity for the pragmatic evaluation of one RIDT within the context of real-life clinical practice where numerous clinicians, at various stages of their training and careers, were engaged to identify suitable ARI patients and collect surveillance data and respiratory specimens.

## Population

Patients of all ages were eligible for inclusion if the clinician identified the presence of an ARI and the patient had at least two acute respiratory tract symptoms (nasal discharge, nasal congestion, sore throat, cough, fever) that began within seven days of their clinic visit. For patients aged ≥2 years, the IISP definition of ILI was fever with cough and/or sore throat [23]. For patients aged <2 years, ILI was defined as fever with ≥1 respiratory symptom(s).

## Procedures

Surveillance program staff provided a brief initial training to all clinicians and trained all incoming family medicine residents. Clinicians collected extensive demographic, epidemiologic, and symptom data (Table 1) on each patient along with paired respiratory specimens: (a) anterior nasal specimen using a nasal swab (Pur-Wraps®) and (b) either a nasopharyngeal (NP) or a high oropharyngeal (OP) specimen using a flocked swab (Copan®). The anterior nasal swab was returned to its paper sheath and immediately transferred to the on-site clinical laboratory. Sofia-FIA was performed on the anterior nasal swab specimen at the time of the patient visit, thus allowing for clinical decision-making. Clinic laboratory technicians followed laboratory-approved procedures for Sofia-FIA, as detailed in the package insert [24]. The NP/OP swab was immediately placed into a labeled 3 ml viral transport medium (Remel MicroTest™ M4RT®) tube, kept at 4–8˚ Celsius, and shipped with a requisition form to the Wisconsin State Lab of Hygiene (WSLH) via courier, usually within 24 hours of collection. The WSLH requisition form contained the Sofia-FIA results, patient demographic information, and clinical information, including number of days from illness onset, symptoms, and whether patients received an influenza vaccine prior to their illness. Surveillance staff confirmed vaccine status through the Wisconsin Immunization Registry [25].

**Table 1. Routinely collected data for patients with acute respiratory infection selected for influenza surveillance at 5 primary care clinics.**

| Data Element | Description |
|---|---|
| **Epidemiological** | |
| Time from symptom onset | Number of days from illness onset to home visit |
| Exposure to similar illness | Exposure to similar illness 1–3 days prior to illness onset |
| **Demographic** | |
| Age | years |
| Sex | male / female / other |
| Race | Standard categories |
| Ethnicity | Standard categories |
| **Clinical** | |
| Illness Severity | Mild, moderate, severe as recorded by clinician |
| Measured temperature | Temperature taken by clinic staff |
| Presence of symptoms | Presence/absence of 17 symptoms; other recorded |
| Use of antipyretics in last 6 hours | Reported by patient |
| Recent influenza antiviral use | Reported by patient |
| Receipt of current seasonal influenza vaccine | Reported by patient (verified through registry) |

## Diagnostics

Clinics transitioned from the Quidel QuickVue® Influenza A+B to Sofia-FIA between October 2012 and April 2013. Both RIDTs are CLIA-waived rapid antigen tests [24]. This change was prompted by a CDC request that Wisconsin clinics serve as a testing site for automated transfer of daily results, as made possible by the wireless feature of Sofia-FIA [26]. The Sofia-FIA and QuickVue platforms have similar specificities, but Sofia-FIA demonstrates superior sensitivity due to immunofluorescence-based lateral-flow technology [27]. The Sofia package insert cites nasal swab sensitivity and specificity as 90% and 95% for influenza A and 89% and 96% for influenza B, respectively [24].

We used the In-vitro Diagnostic (IVD) CDC Human Influenza Virus RT-PCR Panel as our comparison standard [28]. This panel allowed for identification of influenza type and subtype. Cycle threshold values were reported for all specimens. All confirmatory testing was performed at the WSLH. In addition, a respiratory pathogen panel (RPP)—identifying 17 viral targets—was performed on all specimens [29].

## Data analysis

Categorical characteristics were described by counts (%) and continuous characteristics were described by mean (sd). The overall sensitivity, specificity, positive predictive value (PPV), and negative predictive value (NPV) were calculated along with Agresti-Coull confidence intervals. Sensitivity and specificity were described when stratified by each categorical variable.

Unadjusted associations with sensitivity and specificity were analyzed using chi-square tests for categorical variables and Mann-Whitney-Wilcoxon tests for numerical variables. Adjusted models were fit for predicting improvements in sensitivity and specificity based on participant gender, presence of an ILI, days from symptom onset, severity (mild, moderate, or severe), vaccination status, co-detection of other pathogens (RPP), season (early, peak, or late), age, presence/absence of each of a number of symptoms, and an interaction term of age and days from onset. Quadratic terms for age were considered but only included if they significantly improved their model based on likelihood ratio tests. Subjects with unknown gender or vaccination were removed from adjusted models.

The adjusted odds ratios, 95% confidence intervals, and their corresponding p-values were reported. Significance was assessed at the alpha = 0.05 level. No corrections were made to unadjusted or adjusted p-values to control for inflated Type 1 error rate. Binomial logistic regression models were used to predict true positives (sensitivity) and true negatives (specificity) from corresponding RT-PCR samples. Data from 1,126 RT-PCR-positive subjects with no missing data were used in the adjusted sensitivity model, and 3,190 RT-PCR-negative subjects in the adjusted specificity model. Statistical analysis was performed with R version 4.02.

## Results

A total of 5,989 respiratory specimens were collected from October 26, 2012 through June 30, 2019. We excluded 1,514 specimens due to one or more of the following criteria: (1) Sofia RIDT was not performed (n = 1,259 RIDT results were obtained with QuickVue); (2) specimen collected >7 days after illness onset (n = 222); and (3) RT-PCR was not performed (n = 33). We analyzed 4,475 paired specimens, of which one pair was missing symptom information so negatives were imputed for the symptoms. A detailed description of how samples were selected for data analysis is provided in Fig 1.

The demographics of the surveillance population were reflective of general primary care populations with a broad range in patient ages (0.0–98.8 years) and a majority of female patients (Table 2). The majority of patients evaluated for ARI met the ILI criteria (57.4%) and presented for care an average of 3.47 days after symptom onset. Patients presented most commonly with cough (83.7%), nasal discharge (75.6%), sore throat (63.4%), and fever (60.6%). The influenza vaccination rate of 46.2% was slightly above state and national averages [30].

### PCR results

Of the 4,475 specimens collected and analyzed, 1,169 (26.1%) were positive by RT-PCR for influenza (Fig 1). Influenza A(H3) was identified in 553 specimens (47.3% of the positives, 12.4% overall prevalence), influenza A(H1) was identified in 318 specimens (27.2% of the positives, 7.1% overall prevalence), and influenza B was identified in 293 specimens (25% of the positives, 6.5% overall prevalence). Unknown or unsubtypeable influenza A strains were detected in 7 specimens. Cycle threshold values were not available for 141 influenza results, and ranged from 14.25 to 37.79.

### RIDT performance characteristics

Sofia-FIA detected the presence of influenza in 874 specimens, 774 of which were confirmed influenza-positive by RT-PCR (PPV 88.6%). Overall sensitivity of Sofia-FIA for influenza A was 66.2% (95% confidence interval: 63.0–69.3), with a specificity of 97.9% (97.4–98.3). For influenza B, sensitivity was also 66.2% (60.6–71.4) while the specificity was 97.4% (96.8–97.8). A summary of Sofia-FIA performance statistics can be found in Table 3.

### Clinical and laboratory predictors

In an unadjusted analysis, greater sensitivity was associated with the following factors: illness meeting ILI criteria, no non-influenza virus co-detection (RPP), no seasonal influenza vaccination, younger age, lower Ct value, fewer days since illness onset, and presence of nasal discharge, nasal congestion, and fever (p < .05, Table 4A and 4B). Factors not significantly associated with sensitivity were sex, influenza type, illness severity, and seasonality (early, peak, and late influenza season) and all other recorded symptoms. None of the factors were significantly associated with specificity.

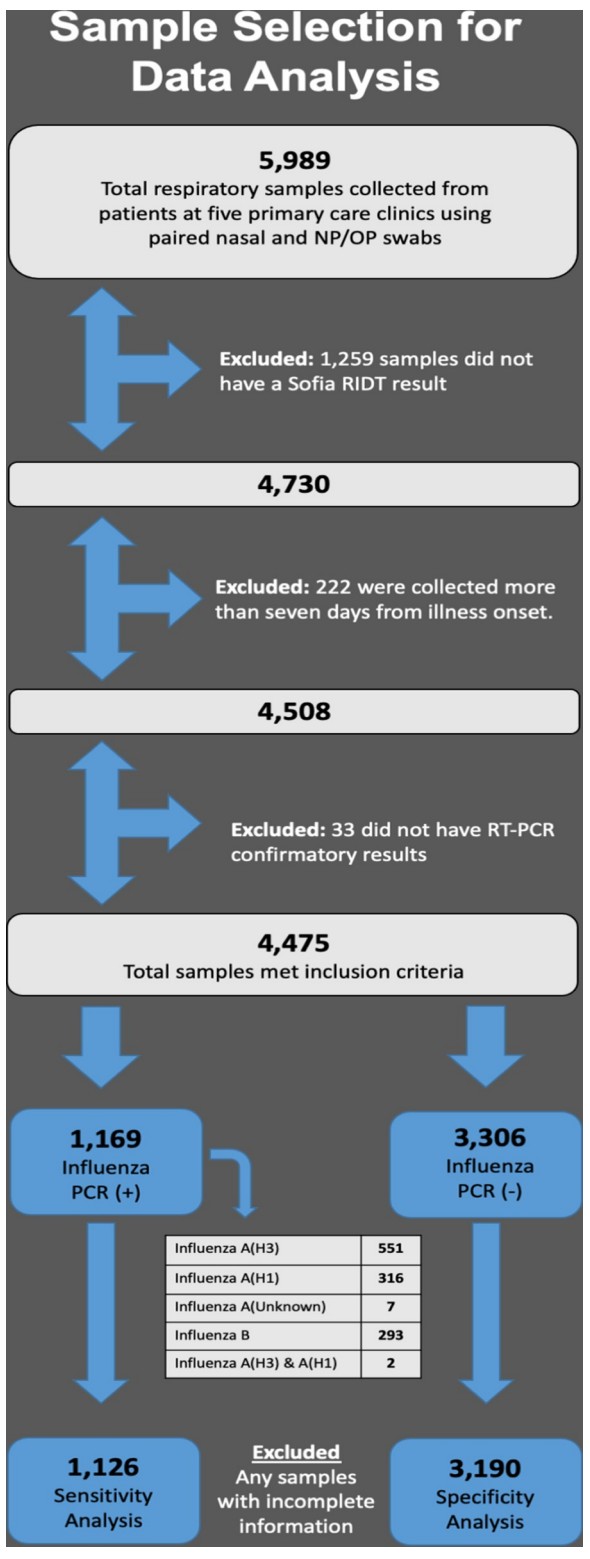

**Fig 1.**

**Table 2. Demographics and distribution of sample characteristics of 4,475 primary care patients presenting with acute respiratory infections and selected for influenza surveillance.**

| Characteristic | Total, n (%) |
|---|---|
| Total specimens | 4,475 |
| Female | 2,687 (60.0) |
| Influenza-like Illness (ILI) | 2,567 (57.4) |
| Vaccinated against influenza | 2,068 (46.2) |
| Days from Onset (mean ± SD) | 3.47 ± 1.79 |
| **Age** | |
| Mean ± SD | 34.9 ± 21.5 |
| Median [range] | 35.08 [.03–98.8] |
| **Clinic** | |
| Belleville (rural) | 837 (18.7) |
| Northeast (urban) | 1,025 (22.9) |
| Oregon (rural) | 360 (8.0) |
| Verona (suburban) | 1,034 (23.1) |
| Wingra (urban) | 1,219 (27.2) |
| **Severity** (as recorded by clinician) | |
| Mild | 1,314 (29.4) |
| Moderate | 2,843 (63.5) |
| Severe | 236 (5.3) |
| **Season** | |
| Early (July-Nov) | 877 (19.6) |
| Peak (Dec-Feb) | 2,224 (49.7) |
| Late (March-June) | 1,374 (30.7) |
| **Symptoms** | |
| Chills | 2,301 (51.4) |
| Cough | 3,746 (83.7) |
| Fever | 2,712 (60.6) |
| Headache | 2,353 (52.6) |
| Malaise | 2,456 (54.9) |
| Myalgia | 1,918 (42.9) |
| Nasal Congestion | 2,750 (61.5) |
| Runny Nose | 3,381 (75.6) |
| Sore Throat | 2,837 (63.4) |
| **PCR Results[a]** | |
| Influenza A H1 | 318 (7.1) |
| Influenza A H3 | 553 (12.4) |
| Influenza A (other) | 7 (0.2) |
| Influenza B | 293 (6.5) |

[a]Both influenza A(H1) and A(H3) were detected in two samples.

**Table 3. Summary performance statistics for Sofia® Influenza A + B fluorescent immunoassay.**

| | Overall | Influenza A | Influenza B |
|---|---|---|---|
| Sensitivity (95% CI) | 66.2 (63.4–68.9) | 66.2 (63.0–69.3) | 66.2 (60.6–71.4) |
| Specificity (95% CI) | 96.2 (95.5–96.8) | 97.9 (97.4–98.3) | 97.4 (96.8–97.8) |
| PPV (95% CI) | 86.1 (83.7–88.2) | 88.5 (85.8–90.7) | 63.7 (58.1–68.9) |
| NPV (95% CI) | 89.0 (87.9–89.9 | 92.2 (91.3–93.0) | 97.6 (97.1–98.0) |

Specificity by type was calculated by including all individuals who did not test positive for a given virus as 'True Negative'.

**Table 4. a: Unadjusted analyses of the effects of clinical and laboratory factors on Sofia® Influenza A + B fluorescent immunoassay sensitivity as compared to RT-PCR.** Subjects were excluded from analysis if necessary data was missing (1 from sex analysis, 24 from severity analysis). Significant results indicated with an asterisk. b: Unadjusted analysis of the effect of age and Ct value and days from illness onset to specimen collection on sensitivity of Sofia® Influenza A + B fluorescent immunoassay. Significant results indicated with an asterisk.

| Characteristic | True Positive | Positive | Sensitivity, n (95% CI) | p-value |
|---|---|---|---|---|
| Male | 350 | 518 | 67.6 (63.4–71.5) | 0.401 |
| Female | 424 | 650 | 65.2 (61.5–68.8) | |
| ILI | 601 | 863 | 69.6 (66.5–72.6) | < 0.001* |
| No ILI | 173 | 306 | 56.5 (50.9–62.0) | |
| Influenza Vaccine | | | | 0.022* |
| Vaccinated | 307 | 490 | 62.7 (58.3–66.8) | |
| Unvaccinated | 456 | 660 | 69.1 (65.5–72.5) | |
| Severity (as recorded by clinician) | | | | 0.973 |
| Mild | 197 | 299 | 65.9 (60.3–71.0) | |
| Moderate | 510 | 767 | 66.5 (63.1–69.7) | |
| Severe | 53 | 79 | 67.1 (56.1–76.5) | |
| Season | | | | 0.403 |
| Early (July-Nov) | 14 | 26 | 53.8 (35.5–71.3) | |
| Peak (Dec-Feb) | 486 | 731 | 66.5 (63.0–69.8) | |
| Late (March-June) | 274 | 412 | 66.5 (61.8–70.9) | |
| Symptom | | | | |
| Chills | 482 | 729 | 66.1 (62.6–69.5) | 0.931 |
| Cough | 723 | 1092 | 66.2 (63.4–69.0) | 0.996 |
| Fever | 615 | 885 | 69.5 (66.4–72.4) | < 0.001* |
| Headache | 449 | 668 | 67.2 (63.6–70.7) | 0.401 |
| Malaise | 463 | 690 | 67.1 (63.5–70.5) | 0.440 |
| Myalgia | 378 | 579 | 65.3 (61.3–69.1) | 0.508 |
| Nasal Congestion | 506 | 737 | 68.7 (65.2–71.9) | 0.021* |
| Runny Nose | 636 | 923 | 68.9 (65.8–71.8) | < 0.001* |
| Sore Throat | 494 | 733 | 67.4 (63.9–70.7) | 0.267 |
| Co-detected viruses | 32 | 60 | 53.3 (40.9–65.4) | 0.031* |
| Single virus detection | 741 | 1108 | 66.9 (64.1–69.6) | |

| Age | True Positive | False Negative | < 0.001* |
|---|---|---|---|
| Mean ± SD | 32.4 ± 21.46 | 39.3 ± 20.13 | |
| Median [IQR] | 33.7 [12.4–49.5] | 41.2 [20.1–54.6] | |
| Ct Value | | | < 0.001* |
| Mean ± SD | 25.3 ± 4.73 | 28.8 ± 4.36 | |
| Median [IQR] | 25.1 [21.8–28.4] | 29.1 [26.0–31.6] | |
| Days from onset | | | < 0.001* |
| Mean ± SD | 2.8 ± 1.52 | 3.3 ± 1.72 | |
| Median [IQR] | 3.0 [2.0–4.0] | 3.0 [2.0–4.0] | |

After adjustment, illness meeting ILI criteria, younger age, fewer days from onset, no co-detection, and presence of a nasal discharge maintained significance. Sensitivity was significantly improved by including a quadratic term for age (LRT p = 0.010). Adjusted odds ratios, confidence intervals, and p-values can be found in Fig 2.

Again, no factors were significantly associated with specificity in the adjusted analysis.

## Discussion

We found that increased sensitivity of Sofia-RIDT was associated with four clinical factors that are readily identifiable at the time of a medical evaluation (presence of an ILI, younger age, fewer days from illness onset, and presence of nasal discharge), and one additional factor that

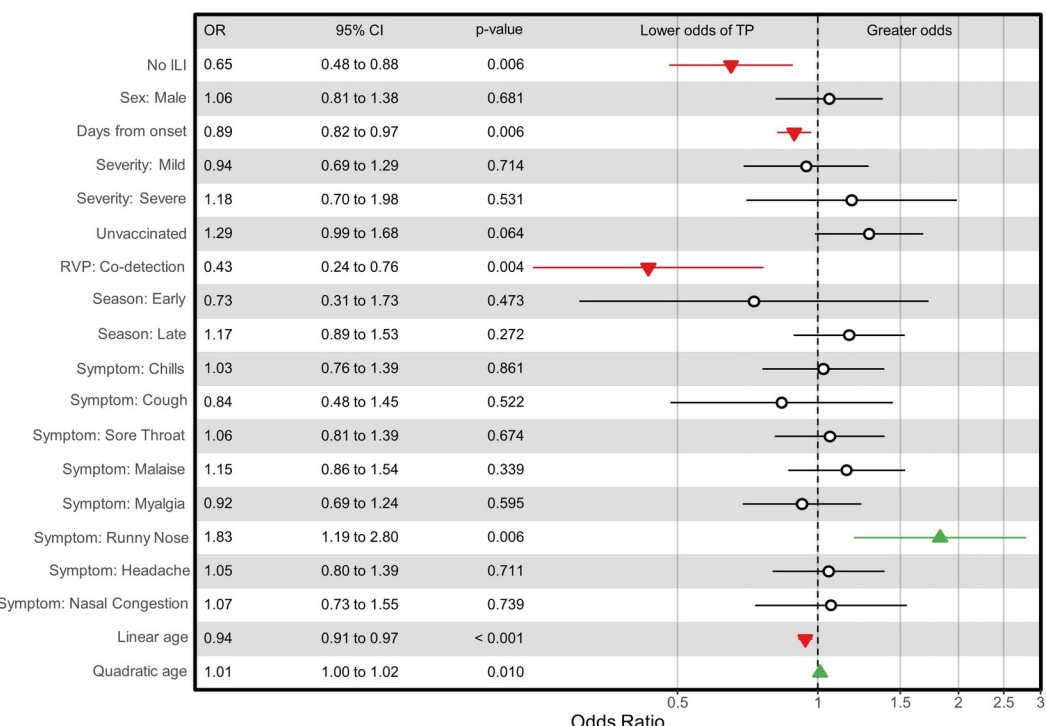

**Fig 2. Odds ratios with 95% confidence intervals for clinical and laboratory factors used in a sensitivity model based on a referent sample from a female patient with an ILI, moderate severity, no flu vaccination, no non-influenza virus co-detection, during peak flu season, and with no symptoms.** Age estimates are per 5 years and centered on the median age for this set (36.1 years). Estimates for days from onset are per 1 additional day. Red triangles depict factors that lower sensitivity. Green triangles depict factors that increase sensitivity.

would not be discernable by a clinician (no co-detection of additional viruses in a respiratory pathogen panel). All of these factors are likely to increase influenza antigenic load in the anterior nares. No factors significantly affected specificity in either unadjusted or adjusted analyses.

Compared with other studies, we had a considerably larger sample size over several consecutive years that included seven sequential influenza seasons. In a 2012 meta-analysis of 159 studies assessing RIDT accuracy, the average sample size was 131 confirmed cases of influenza [31]. Only two of the studies referenced had over 1,000 influenza positive samples, and all samples were collected during a single influenza season, possibly limiting generalizability [32,33]. In addition to incorporating multiple influenza seasons that differed in timing, intensity, and predominant types/subtypes, this study introduced additional variability by including dozens of clinicians who had received the level of training for respiratory specimen collection typical in primary care settings. Accordingly, the results from this pragmatic assessment are more likely to represent performance of Sofia-FIA in real-life settings.

Many studies have noted the effect of age on RIDT sensitivity [6,12–19], but few have considered other clinical factors that could contribute to RIDT performance. More commonly, studies examine laboratory factors such as Ct value and virus type and subtype, which are not generally available during a clinical session. Our study assessed simultaneously eight potential clinical factors (ILI status, sex, age, illness severity, common respiratory symptoms, seasonality, influenza vaccination status, and days from illness onset) and three laboratory factors (Ct value, influenza type, and detection of another virus).

The effect of Ct value, sex, age, and time from illness onset on RIDT performance are consistent with previous findings within the literature [17,34]. Seasonality is not well defined, but studies indicate that RIDTs are most useful when community prevalence of influenza is high because positive predictive value is greatest at that time [10]. We are unaware of any studies that assess the effect of vaccination status and individual symptoms.

Few studies have examined the relationship between sensitivity and the presence/absence of an ILI. During a performance assessment of QuickVue, Koul et al. reported no difference in sensitivity for patients with an ILI or a severe acute respiratory illness (SARI) [16]. ILI is well defined in the literature, but how severity is measured for sensitivity analysis varies greatly. SARI was defined as those who have ILI (fever accompanied by cough and/or sore throat) and are hospitalized. Another study that used hospitalization as a marker for severity found that sensitivity was especially poor among hospitalized adults (45%) compared with outpatient adults (75%), with a similar mean time from illness onset (2.7 days and 2.1 days, respectively) [35]. Hospitalized children had a higher sensitivity (84%), which was likely due to the higher viral loads commonly found in younger populations. The higher sensitivity among children may be due to a higher viral load in younger populations. In our study, we did not implement an age cutoff, and sensitivity was greater for those who had an ILI, but it was not influenced by severity. Our definition of severity was based on a clinician-reported three-point scale and may be subject to bias. A more uniform definition of severity may be needed.

As is commonly found in the scientific literature, Sofia-FIA did not perform as well as expected in real-world clinical settings. The Sofia package insert cites nasal swab sensitivity and specificity as 90% and 95% for influenza A and 89% and 96% for influenza B, respectively [24]. In contrast, we found an overall sensitivity for influenza A of 66.2% (95% confidence interval: 63.0–69.3), with a specificity of 97.9% (97.4–98.3). For influenza B, sensitivity was also 66.2% (60.6–71.4) while the specificity was 97.4% (96.8–97.8). Most studies have indicated lower sensitivity for influenza B, but our study showed 66.2% sensitivity for both influenza A and B [19,22,36]. Furthermore, our study found that detecting another virus could decrease sensitivity. Although research in this area is limited, a study comparing four RIDTs saw no cross reactivity in samples with co-infections [37].

This analysis had several limitations. First, we only analyzed the performance of one RIDT, but many of our findings are compatible to those of studies of other RIDTs. Second, our surveillance program encompasses a limited geographical area in Southcentral Wisconsin, and we defined seasonality based on the temperate climate. The distribution of influenza types and subtypes, however, are similar to distributions reported nationally in any given year. For the purpose of generalizability, we defined seasonality as early (July-November), peak (December-February), and late (March-June) for each year analyzed, but influenza outbreaks occur at varying times of the year, and this may have had an effect on the analysis. Third, by utilizing multiple clinicians to identify potential patients, collect clinical data, and obtain respiratory specimens, much more variability is introduced than would occur in other research settings. This study provides results obtained in real-world primary care settings in which patient selection, specimen collection, and testing occurred within routine clinical activities. Although this is an asset in determining whether RIDTs are appropriate for point-of-care diagnosis and treatment, we cannot guarantee rigor in all data elements to the same degree as a more traditionally controlled research study or randomized clinical trial. Moreover, variability in patient selection and specimen collection may contribute to lower estimates of sensitivity. Finally, despite the large sample size, there may not have been sufficient power to identify factors that may affect specificity. Their effects, however, would be trivial given the overall high specificity.

Several clinical and laboratory factors in this study appear to affect RIDT sensitivity. Awareness of these factors and their identification and consideration by clinicians at the point-of-

care may aid in patient selection and the appropriate interpretation of negative influenza RIDT results. If a clinician suspects that a patient has influenza, but an RIDT test is negative, the clinician may be able to take age, symptoms, and days from illness onset into consideration when assessing whether or not the results are accurate.

Due to the ongoing SARS-CoV-2 pandemic, clinicians are relying more on diagnostic tests than symptom assessment, and nucleic acid amplification testing (NAAT) has become more widely available in multiple areas of the world. NAAT tests are sensitive and specific, but they can be costly and it can take hours to days to receive results. RIDTs, in contrast, are relatively inexpensive and produce results in the time that it takes a clinician to assess and treat patients. Thus, an RIDT can be used as an initial tool during point-of-care with the caveat that some factors my influence sensitivity and NAAT confirmation may be necessary.

Additional performance characteristic studies analyzing potential factors affecting sensitivity are necessary to explain broad ranges sensitivity across RIDT platforms and to establish standard guidelines for clinical interpretation of influenza RIDT results.

## Acknowledgments

We would like to acknowledge Dr. Mindy Smith for her role in providing critical feedback on this manuscript.

## Author Contributions

**Conceptualization:** Cristalyne Bell, Maureen Goss, Emily Temte, Peter Shult, Erik Reisdorf, Thomas Haupt, Shari Barlow, Jonathan Temte.

**Data curation:** Cristalyne Bell, Maureen Goss, Emily Temte, Shari Barlow, Jonathan Temte.

**Formal analysis:** Jennifer Birstler, Guanhua Chen.

**Funding acquisition:** Shari Barlow, Jonathan Temte.

**Investigation:** Cristalyne Bell, Peter Shult, Erik Reisdorf, Thomas Haupt, Jonathan Temte.

**Methodology:** Jonathan Temte.

**Project administration:** Cristalyne Bell, Maureen Goss, Emily Temte, Shari Barlow, Jonathan Temte.

**Resources:** Peter Shult, Erik Reisdorf, Thomas Haupt.

**Supervision:** Shari Barlow, Jonathan Temte.

**Validation:** Emily Temte.

**Visualization:** Jennifer Birstler.

**Writing – original draft:** Cristalyne Bell, Maureen Goss, Jennifer Birstler, Jonathan Temte.

**Writing – review & editing:** Cristalyne Bell, Maureen Goss, Jennifer Birstler, Emily Temte, Guanhua Chen, Erik Reisdorf, Thomas Haupt, Shari Barlow, Jonathan Temte.

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
