## [Decision Letter · Decision Letter 0]

9 Nov 2021

PONE-D-21-09174

Assessment of potential factors associated with the sensitivity and specificity of Sofia Influenza A+B Fluorescent Immunoassay in ambulatory care

PLOS ONE

Dear Dr. Bell,

Thank you for submitting your manuscript to PLOS ONE. After careful consideration, we feel that it has merit but does not fully meet PLOS ONE’s publication criteria as it currently stands. Therefore, we invite you to submit a revised version of the manuscript that addresses the points raised during the review process.

The manuscript has been evaluated by two reviewers, and their comments are available below.

The reviewers have raised a number of concerns that need attention. They request additional information details to improve the quality of the reporting of the study, including the statistical analyses.

Could you please revise the manuscript to carefully address the concerns raised?

We look forward to receiving your revised manuscript.

Kind regards,

Marianne Clemence

Associate Editor

PLOS ONE

Journal Requirements:

2. Please provide the full names of the five primary care clinics in southcentral Wisconsin.

"JLT has received past research funding from Quidel Corporation: https://www.quidel.com/. Quidel has provided Sofia analyzers and Influenza A+B FIA tests to the Wisconsin surveillance team. Quidel did not play a role in study design, data collection and analysis, decision to publish, or preparation of the manuscript"

We note that you received funding from a commercial source: Quidel Corporation

Reviewers' comments:

Reviewer's Responses to Questions

**Comments to the Author**

1. Is the manuscript technically sound, and do the data support the conclusions?

Reviewer #1: Yes

Reviewer #2: Yes

2. Has the statistical analysis been performed appropriately and rigorously? 

Reviewer #1: I Don't Know

Reviewer #2: Yes

3. Have the authors made all data underlying the findings in their manuscript fully available?

Reviewer #1: Yes

Reviewer #2: No

4. Is the manuscript presented in an intelligible fashion and written in standard English?

Reviewer #1: Yes

Reviewer #2: Yes

5. Review Comments to the Author

Reviewer #1: The submitted manuscript of Bell et al. assesses potential clinical and laboratory factors which are associated with the sensitivity and specificity of Sofia Influenza A+B fluorescent immunoassay. Those immune assays are easy to use and provide results within a short time frame. As pointed out and statistically tested by Bell et. al. caution is required since the sensitivity is extremely low. They found 4 out of 11 tested factors that can influence the results and should be monitored in future. All of these factors are likely to increase antigenic load. Although the results reach statistical significance for those 4 factors, the sensitivity does not increase higher than 70%. In conclusion, those four factors are helpful for future diagnosis nevertheless those factors are no game changer nor is it known how these results could be implemented to increase the accuracy in near future. The following minor concerns might improve the quality of the manuscript and strengthen their findings

- Flow diagram of sample selection would be very helpful

- What does “a quadratic term of age” mean?

- One would expect increased sensitivity in severe cases!? (higher AG load!?!)

- Since ct and “days from illness onset” are comprehensible to reach statistical significance, why does the little mean age difference (32.4 / 39.3) reached statistical difference in your analysis?

Reviewer #2: The strength of this paper is the testing over multiple years and the number of tests performed. The identified risk factors for a positive RAT are not surprising, but it is important to publish these data.

The paper is well-written, but a number of questions arises, which I hope the authors are able to clarify.

In general, children and adults are normale separated as younger people are more likely to have symptoms such as fever and to be positive for multiple respiratory pathogens at a time, whereas elderly people often doesn't get fever and only are positive for one pathogen at a time. Please consider that you may be comparing two different patient populations in your statistical analysis, and should perform the analysis separately for children and adults.

On page 9 top. The IFU sensitivity and specificity is reported and the authors calculate a clinical sensitivity and specificity (RIDT performance on page 12), but does not include these data in the discussion. Please reflect on your findings compared to the reported values in the IFU in the discussion, as these data are interesting.

On page 10 second paragraph. Data from 1126 RT-PCR positive is included together with 3190 RT-PCR negative or in total 4316 individuals, but in the next paragraph (results), 4475 paired specimens are included. Is this difference due to multiple testing and if so, then please report the number of repeated testing to allow the reader to understand the math of calculations.

On page 12 PCR results: 1169 were positive by RT-PCR (how is this possible when 1126 were RT-PCR positive) and 1171 are reported positive in table 2, so are two individuals dual-positives?

On page 13 Table 4: Please define severity in the manuscript, what is mild/moderate/severe? and please define the season, what is early/peak and late, does this change from year to year or do you use fixed periods of time?

In the discussion. Please comment on the role of RAT, when bearing in mind that NAAT based platforms e.g. Liat, ID now, biofire or QIAstat are now commonly available in multiple areas of the world.

6. PLOS authors have the option to publish the peer review history of their article (what does this mean?). If published, this will include your full peer review and any attached files.

Reviewer #1: No

Reviewer #2: **Yes: **Uffe Vest Schneider

---

## [Author Response · Author response to Decision Letter 0]

24 Dec 2021

JOURNAL REQUIREMENTS

I have formatted the manuscript per the guidelines listed in the links you provided above. 

2. Please provide the full names of the five primary care clinics in southcentral Wisconsin.

• UW Health Belleville Family Medicine

• UW Health Verona Clinic

• UW Health Oregon Clinic

• UW Health Northeast Family Medical Center

• UW Health Wingra Family Medical Center

We did not receive additional funding for this analysis, but I have updated the “Funding Information” with the grant award number: MSN256387. I have also deleted the “Financial Disclosure” in the manuscript, per your online instructions: https://journals.plos.org/plosone/s/submission-guidelines#loc-references

"JLT has received past research funding from Quidel Corporation: https://www.quidel.com/. Quidel has provided Sofia analyzers and Influenza A+B FIA tests to the Wisconsin surveillance team. Quidel did not play a role in study design, data collection and analysis, decision to publish, or preparation of the manuscript"

We note that you received funding from a commercial source: Quidel Corporation

I amended the Competing Interest Statement in the manuscript to explicitly state that Quidel Corporation provided “in-kind” materials. I also stated that this does not alter our ability to adhere to all PLOS ONE policies on sharing data and materials. The statement now reads as follows: 

JLT has received past research funding from Quidel Corporation. Quidel provided in-kind Sofia analyzers and Influenza A+B FIA tests to the Wisconsin surveillance team. This does not alter our adherence to PLOS ONE policies on sharing data and materials. Quidel did not direct or exert any influence over study design, data collection and analysis, decision to publish, or preparation of the manuscript.

REVIEWER COMMENTS

1. Is the manuscript technically sound, and do the data support the conclusions?

Reviewer #1: Yes

Reviewer #2: Yes

Thank you for reviewing our manuscript. We are pleased to know that you feel our manuscript is technically sound and that our conclusions are supported by the data. 

2. Has the statistical analysis been performed appropriately and rigorously? 

Reviewer #1: I Don't Know

Reviewer #2: Yes

The statistical analysis was performed with the oversight of a PhD-level biostatistician. 

3. Have the authors made all data underlying the findings in their manuscript fully available?

Reviewer #1: Yes

Reviewer #2: No

Dataset can be accessed at the following data repository: https://dataverse.harvard.edu/dataset.xhtml?persistentId=doi:10.7910/DVN/JNC1XJ

4. Is the manuscript presented in an intelligible fashion and written in standard English?

Reviewer #1: Yes

Reviewer #2: Yes

We are pleased that the quality of writing in our manuscript meets your standards. 

5. Review Comments to the Author

Reviewer #1

The submitted manuscript of Bell et al. assesses potential clinical and laboratory factors which are associated with the sensitivity and specificity of Sofia Influenza A+B fluorescent immunoassay. Those immune assays are easy to use and provide results within a short time frame. As pointed out and statistically tested by Bell et. al. caution is required since the sensitivity is extremely low. They found 4 out of 11 tested factors that can influence the results and should be monitored in future. All of these factors are likely to increase antigenic load. Although the results reach statistical significance for those 4 factors, the sensitivity does not increase higher than 70%. In conclusion, those four factors are helpful for future diagnosis nevertheless those factors are no game changer nor is it known how these results could be implemented to increase the accuracy in near future. The following minor concerns might improve the quality of the manuscript and strengthen their findings

- Flow diagram of sample selection would be very helpful

We have provided a flow chart to demonstrates how the samples were selected for analysis. 

- What does “a quadratic term of age” mean?

Quadratic term of age is age-squared and implies that the fit is better with a non-linear relationship. This is fairly established in the literature. 

- One would expect increased sensitivity in severe cases!? (higher AG load!?!)

This is not what we found in our analysis. How severity is defined may be a factor. In other studies, severe cases are equated with hospitalization. In our case, clinicians report severity based on a 3-point scale. Both definitions may be subject to bias, ours is perhaps even more so. We did, however, see a relationship between sensitivity and the presence of an influenza-like illness (ILI). The definition of ILI is based on whether a patient has specific symptoms. A yes or no to the presence of a symptom may be a better measure than severity in this case. We had added the following information and an additional citation to clarify that the definition of severity varies within the literature. 

“ILI is well defined in the literature, but how severity is measured for sensitivity analysis varies greatly. SARI was defined as those who have ILI (fever accompanied by cough and/or sore throat) and are hospitalized. Another study that used hospitalization as a marker for severity found that sensitivity was especially poor among hospitalized adults (45%) compared with outpatient adults (75%), with a similar mean time from illness onset (2.7 days and 2.1 days, respectively).[35] Hospitalized children had a higher sensitivity (84%), which was likely due to the higher viral loads commonly found in younger populations. In our study, we did not implement an age cutoff, and sensitivity was greater for those who had an ILI, but it was not influenced by severity. Our definition of severity was based on a clinician-reported three-point scale and may be subject to bias. A more uniform definition of severity may be needed.

- Since ct and “days from illness onset” are comprehensible to reach statistical significance, why does the little mean age difference (32.4 / 39.3) reached statistical difference in your analysis?

The mean age is consistent with what we would expect to see in primary care clinics. Cycle threshold, days from illness onset, and age were all significant when unadjusted. Age was still significant in the adjusted analysis. 

 

Reviewer #2

The strength of this paper is the testing over multiple years and the number of tests performed. The identified risk factors for a positive RAT are not surprising, but it is important to publish these data.

The paper is well-written, but a number of questions arises, which I hope the authors are able to clarify.

In general, children and adults are normale separated as younger people are more likely to have symptoms such as fever and to be positive for multiple respiratory pathogens at a time, whereas elderly people often doesn't get fever and only are positive for one pathogen at a time. Please consider that you may be comparing two different patient populations in your statistical analysis, and should perform the analysis separately for children and adults.

It is common in the literature to divide patient population into pediatric and adult cases, but we decided against implementing an age cutoff and instead accounted for age in the multivariant analysis. 

On page 9 top. The IFU sensitivity and specificity is reported and the authors calculate a clinical sensitivity and specificity (RIDT performance on page 12), but does not include these data in the discussion. Please reflect on your findings compared to the reported values in the IFU in the discussion, as these data are interesting.

We have added the following paragraph to the Discussion section: 

As is commonly found in the scientific literature, Sofia-FIA did not perform as well as expected in real-world clinical settings. The Sofia package insert cites nasal swab sensitivity and specificity as 90% and 95% for influenza A and 89% and 96% for influenza B, respectively.(24) In contrast, we found an overall sensitivity for influenza A of 66.2% (95% confidence interval: 63.0—69.3), with a specificity of 97.9% (97.4—98.3). For influenza B, sensitivity was also 66.2% (60.6—71.4) while the specificity was 97.4% (96.8—97.8).

On page 10 second paragraph. Data from 1126 RT-PCR positive is included together with 3190 RT-PCR negative or in total 4316 individuals, but in the next paragraph (results), 4475 paired specimens are included. Is this difference due to multiple testing and if so, then please report the number of repeated testing to allow the reader to understand the math of calculations.

On page 12 PCR results: 1169 were positive by RT-PCR (how is this possible when 1126 were RT-PCR positive) and 1171 are reported positive in table 2, so are two individuals dual-positives?

We apologize for the confusion. To help clarify how we selected samples for analysis, we have created a flow chart (Fig 1) and added a note to table two about co-detections. 

On page 13 Table 4: Please define severity in the manuscript, what is mild/moderate/severe? and please define the season, what is early/peak and late, does this change from year to year or do you use fixed periods of time?

We defined season in Table 4 as Early (July-November), Peak (December-February), Late (March-June) so that it matches Table 2 and added “as reported by clinician” to Table 2 and Table 4 to match Table 1. As stated in the discussion, seasonality is not well defined in the literature. We added the additional sentence below to the limitations: 

For the purpose of generalizability, we defined seasonality as early (July-November), peak (December-February), and late (March-June) for each year analyzed, but influenza outbreaks occur at varying times of the year and this may have had an effect on the analysis. 

In the discussion. Please comment on the role of RAT, when bearing in mind that NAAT based platforms e.g. Liat, ID now, biofire or QIAstat are now commonly available in multiple areas of the world.

The following paragraph has been added to the discussion section:

Due to the ongoing SARS-CoV-2 pandemic, clinicians are relying more on diagnostic tests than symptom assessment, and nucleic acid amplification testing (NAAT) has become more widely available in multiple areas of the world. NAAT tests are sensitive and specific, but can be costly and it can take hours to days to receive results. RIDTS, in contrast, are relatively inexpensive and produce results in the time that it takes a clinician to assess and treat patients. Thus, an RIDT can be used as an initial tool during point-of-care with the caveat that some factors my influence sensitivity and NAAT confirmation may be necessary.

---

## [Decision Letter · Decision Letter 1]

27 Apr 2022

Assessment of potential factors associated with the sensitivity and specificity of Sofia Influenza A+B Fluorescent Immunoassay in an ambulatory care setting

PONE-D-21-09174R1

Dear Dr. Bell,

Thank you for submitting your manuscript to PLOS ONE; I sincerely apologise for the unusually delayed review timeframe. We’re pleased to inform you that your manuscript has been judged scientifically suitable for publication and will be formally accepted for publication once it meets all outstanding technical requirements.

Kind regards,

Emily Chenette

Editor in Chief

PLOS ONE

Additional Editor Comments (optional):

Reviewers' comments:

Reviewer's Responses to Questions

**Comments to the Author**

1. If the authors have adequately addressed your comments raised in a previous round of review and you feel that this manuscript is now acceptable for publication, you may indicate that here to bypass the “Comments to the Author” section, enter your conflict of interest statement in the “Confidential to Editor” section, and submit your "Accept" recommendation.

Reviewer #1: All comments have been addressed

2. Is the manuscript technically sound, and do the data support the conclusions?

Reviewer #1: Yes

3. Has the statistical analysis been performed appropriately and rigorously? 

Reviewer #1: Yes

4. Have the authors made all data underlying the findings in their manuscript fully available?

Reviewer #1: Yes

5. Is the manuscript presented in an intelligible fashion and written in standard English?

Reviewer #1: Yes

6. Review Comments to the Author

Reviewer #1: In their recently revised manuscript, Bell et al. have managed to largely address the concerns and suggestions that I raised during the initial round of review.

7. PLOS authors have the option to publish the peer review history of their article (what does this mean?). If published, this will include your full peer review and any attached files.

Reviewer #1: No

---

## [Editor Report · Acceptance letter]

1 May 2022

PONE-D-21-09174R1 

Assessment of potential factors associated with the sensitivity and specificity of Sofia Influenza A+B Fluorescent Immunoassay in an ambulatory care setting 

Dear Dr. Bell:

I'm pleased to inform you that your manuscript has been deemed suitable for publication in PLOS ONE. Congratulations! Your manuscript is now with our production department. 

Kind regards, 

on behalf of

Dr Emily Chenette 

Staff Editor

PLOS ONE